# Learning Rotation-Agnostic Representations via Group Equivariant VAEs

**Ahmedeo Shokry**     **Antonio Norelli**

Physics & CS department at La Sapienza University of Rome, Italy

## Abstract

An emerging field in representation learning involves the study of group-equivariant neural networks, that leverage concepts from group representation theory to design neural architectures that can exploit discrete and continuous symmetries to produce more general representations. Following this direction, in this work we demonstrate how an image embedding agnostic to rotations can be naturally obtained by training a variational autoencoder ($\mathcal{S}$-GVAE) equipped with a Group equivariant Convolutional Neural Network (G-CNN) encoder[1].

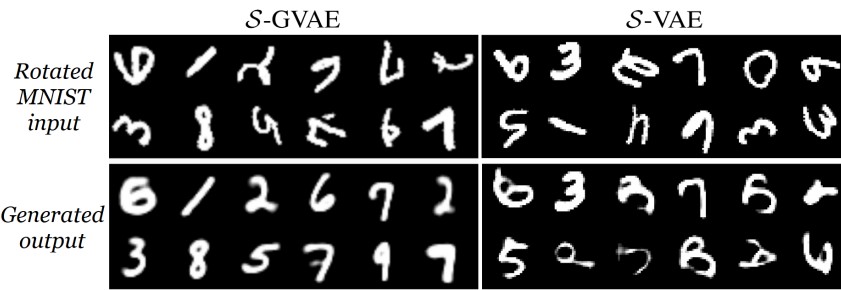

Figure 1: Rotated MNIST reconstructions of a $\mathcal{S}$-GVAE and a standard $\mathcal{S}$-VAE.

## 1 Introduction and related work

Group equivariant Convolutional Neural Networks (G-CNNs) are a generalization of regular CNNs (LeCun & Bengio, 1995) that can exploit symmetries by sharing more weights. They have been shown to outperform regular CNNs on rotated and intrinsically symmetric data, increasing the expressive capacity of the network (Cohen & Welling, 2016a). Recent theoretical work has expanded the capabilities of G-CNNs, which can now deal with any compact continuous group transformation, e.g. continuous image rotations (Weiler & Cesa, 2021; Cesa et al., 2022).

Most of the existing work on G-CNNs has focused on classification tasks (Weiler & Cesa, 2021). In this work, instead, we will present some interesting results obtained by implementing a $SO(2)$ G-CNN inside a VAE (Kingma & Welling, 2013), in order to disentangle the image content from its rotation. Image representations agnostic to rotations are desirable in several scenarios, such as when working with astronomical data or histopathological images (Veeling et al., 2018). Recent works have explored ways to disentangle pose features, such as in Bepler et al. (2019), where the generative part of the autoencoder is an explicit function of the spatial coordinates of the image, enabling the model to factorize and discard the underlying rotation and translation. Vadgama et al. (2022) demonstrated the possibility of creating a VAE that not only disentangles the latent space but also learns to compress information into more straightforward geometric symbols. Similar to our work, they constrained the latent space to assume a hyperspherical manifold and utilized G-CNN to develop a fully equivariant VAE, illustrating how equivalence classes correspond to invariant shapes. However, we will soon explore an alternative approach where we directly implement a $SO(2)$ equivariant CNN as the encoder of a hyperspherical $\beta$-VAE (Davidson et al., 2018; Higgins et al., 2017, $\mathcal{S}$-VAEs), which inherently generates representations independent of rotations. Additionally, we will show that our model produces generated images that can be classified with higher accuracy compared to those produced by a non-equivariant VAE.

---

[1] All the code needed to reproduce this work is available on Colab SGVAE and SVAE with test ResNet-50

## 2 METHODS

Regular group convolutions are an extension of the classical CNNs, where input images and feature maps are modeled as functions $f : \mathbb{Z}^2 \longrightarrow \mathbb{R}^K$. Here $\mathbb{Z}^2$ is the group containing spatial coordinates $(x, y)$ and $K$ is the number of channels. Instead, in a G-CNN the feature maps are functions on a generic group $G$, therefore the notion of convolution can be generalized so that the *shift* becomes a more general transformation in $G$ (Cohen & Welling, 2016a). To make this idea work with continuous groups and e.g. build a G-CNN encoder invariant to rotations, we impose the *steerability* property of the feature field (Cohen & Welling, 2016b; Weiler & Cesa, 2021), see Appendix.

## 3 EXPERIMENTAL RESULTS: ARE 6S JUST ROTATED 9S?

We compared our $\mathcal{S}$-GVAE with a standard $\mathcal{S}$-VAE of the same size ($\sim 800k$ params), both trained on non-rotated MNIST (LeCun et al., 1998). Figure 1 shows the reconstructions of rotated digits from MNIST, as we see the $\mathcal{S}$-GVAE ignores the rotation and reconstructs properly the digits. To evaluate the reconstructions quantitatively, we used a pre-trained ResNet50 (He et al., 2016) to classify the generated outputs from the two models. We also tested their capabilities in reconstructing 6 and 9 digits, which can be easily confused for a rotational equivariant encoder. The results are displayed below.

Table 1: ResNet50 accuracy on MNIST reconstructions from the $\mathcal{S}$-GVAE and $\mathcal{S}$-VAE models.

| input | $\mathcal{S}$-GVAE | $\mathcal{S}$-VAE | input | $\mathcal{S}$-GVAE | $\mathcal{S}$-VAE |
|---|---|---|---|---|---|
| MNIST | 91% | **97%** | Rotated 6 digits | **95%** | 24% |
| Rotated MNIST | **79%** | 32% | Rotated 9 digits | **66%** | 28% |

As expected, the $\mathcal{S}$-GVAE largely outperforms the $\mathcal{S}$-VAE when MNIST images are rotated. On the other hand, the $\mathcal{S}$-GVAE falls short of the $\mathcal{S}$-VAE on standard MNIST, since rotation is sometimes a useful feature for the sake of classification, but the $\mathcal{S}$-GVAE is blind to rotation. Still, the high performance on the sole 6s suggests that the rotation feature is not essential, and therefore that 6s are not just rotated 9s, as evidenced in Figure 2. Plausibly, the discrepancy between the 6 and 9-digit performance derives from the different writing styles. While the 6-digit is consistent across the dataset, a single curve with a curl; the writing style of the 9-digit can vary greatly, with some resembling the traces of 4 or 3-digits, and therefore more difficult to classify.

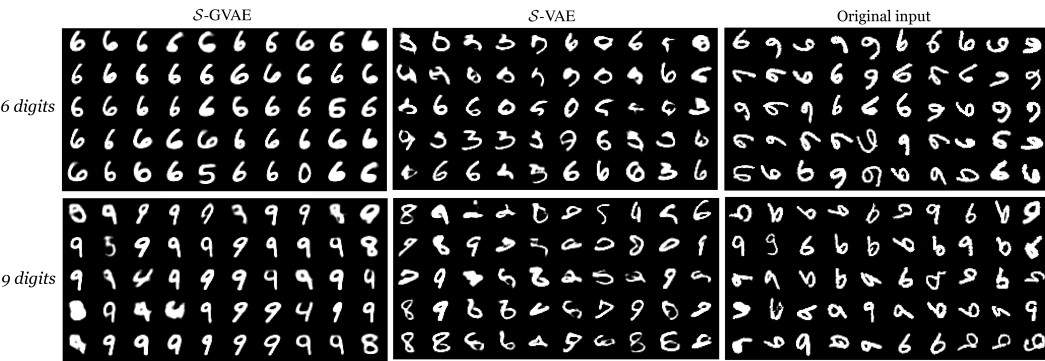

Figure 2: Rotational disentanglement of 6 and 9 digits using a $\mathcal{S}$-GVAE and a $\mathcal{S}$-VAE.

## 4 CONCLUSIONS

In this work, we have presented $\mathcal{S}$-GVAE, a Group Variational Autoencoder (VAE) that incorporates a hyperspherical latent space and a Group equivariant encoder. The primary objective is to achieve rotational disentanglement with minimal effort. By leveraging the rotational symmetry of a G-CNN as an encoder prior, we create a highly expressive VAE that not only generates rotation-agnostic representations but also demonstrates superior performance in terms of reconstruction, showcasing the superior generalization ability of such models. The conducted tests provide a good basis for future experiments and pave the way for further enhancements.

URM STATEMENT

The first author meets the criteria of the ICLR 2023 Tiny Papers Track.

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

## A APPENDIX

In this appendix we will give all the details useful to reproduce the experiments of this work. Starting from the architecture, we used a classical VAE with the addition of a regularization element such as a hyperspherical latent space, and the replacement of the classical encoder with a group equivariant one.

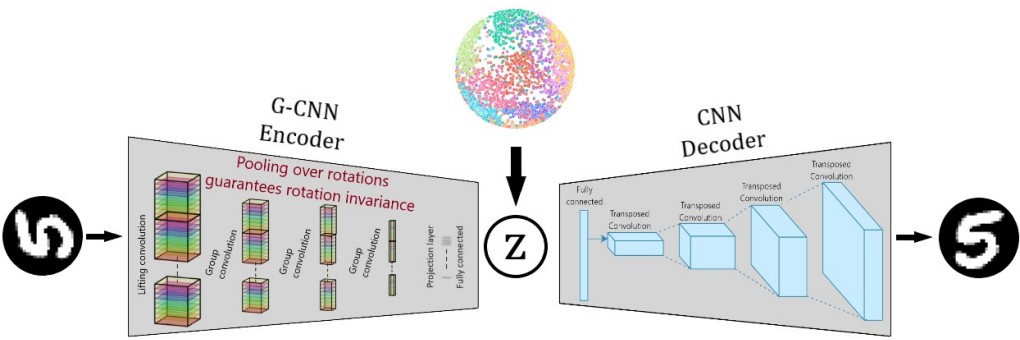

Figure 3: Sketch of the $\mathcal{S}$-GVAE: The architecture is composed by a group equivariant decoder in order to ensure rotation invariance and a hyperspherical latent space as a regularization element[3].

The main purpose of using a group equivariant encoder is to exploit the group simmetry in order to make the $\mathcal{S}$-VAE invariant to rotation. A Hyperspherical Group equivariant Variational Autoencoder ($\mathcal{S}$-GVAE) is composed of an encoder and a decoder, like any other VAE. The only difference is in the encoder part, which is now composed by a G-CNN. The objective here is always to optimize a loss function:

$$\mathcal{L}(\phi, \psi) = \mathbb{E}_{q_\psi(\mathbf{z}|\mathbf{x};\theta)}[\log p_\phi(\mathbf{x}|\mathbf{z})] - \beta \cdot KL(q_\psi(\mathbf{z}|\mathbf{x};\theta)||p(\mathbf{z}))$$

Where $\mathcal{L}$ is the function to be maximized and the right end side is called Evidence Lower Bound (ELBO). In this experiment the approximate posterior is given by the von Mises-Fisher distribution and the probability distribution $p(\mathbf{z})$ of the latent space is the hypersferical uniform distribution, so that the $KL$ divergence in the loss function is $KL(\text{vMF}(\mu, \kappa)||U(S^{n-1}))$ where $\mu$ is the mean value, $\kappa$ is the concentration and $S^{n-1}$ is the unit sphere of dimension $n-1$ where $n$ is the latent dimension.

Now in details, the architecture used for the experiment:

**Dataset**:

MNIST subset of 35000 elements randomly rotated by $\theta \in [-180, 180]$ degree. The split between train and test set was stratified with test set = 1/7 of the entire dataset.

**Optimizer**:

Adam optimizer with initial learning rate of 1e-2, weight decay of 1e-5 and exponential scheduler with $\gamma = 0.9$.

**Architecture $\mathcal{S}$-GVAE**: $\sim 800k$ params

*Input*: Batch (100 element) of 32x32x1 MNIST data (pad = 4).

---

[3]G-CNN encoder illustration adopted from the course "Deep Learning 2: Group equivariant deep learning" `https://uvagedl.github.io/`, University of Amsterdam.

*Encoder*: The encoder is made of an initial mask module, a FourierELU activation (frequency = 6), 3 G-steerable convolutional layers with 64, 64, 128 channels, stride = 2, 1, 2, kernel size = 4, FourierElu activation with frequency = 6, BatchNorm and Dropout = 0.1 each. A pointwise average pooling with stride = 1 was applied to the last layer. Then a G-convolution with kernel = 1 (to extract the invariant feature maps) and two last FC layers for the mean ($\mu$) and concentration ($\kappa$) values of the von Mises-Fisher distribution with ELU activation, BatchNorm and Dropout = 0.2.

*Latent*: Latent dimension set to 3, 6, 10 in order to test different configurations.

*Decoder*: The decoder is made of a first FC linear layer with ReLU activation, BatchNorm and Dropout = 0.1, then 3 Transposed Convolutional layers with 128, 64, 32 channels, stride = 2, 1, 2, kernel size = 4, ReLU activation, BatchNorm and Dropout = 0.2 each.

**Architecture $\mathcal{S}$-VAE**: $\sim$ 800k params

*Input*: Batch (100 element) of 32x32x1 MNIST data (pad = 4).

*Encoder*: The encoder is made of 3 Convolutional layers with 128, 128, 64 channels, stride = 2, 1, 2, kernel size = 4, ReLU activation, BatchNorm and Dropout = 0.2 each. Then two last FC layers for the mean ($\mu$) and concentration ($\kappa$) values of the von Mises-Fisher distribution with ReLU activation, BatchNorm and Dropout = 0.1.

*Latent*: Latent dimension set to 3, 6, 10 as for the $\mathcal{S}$-GVAE.

*Decoder*: The decoder is made of a first FC linear layer with ReLU activation, BatchNorm and Dropout = 0.1, then 3 Transposed Convolutional layers with 128, 128, 64 channels, stride = 2, 1, 2, kernel size = 4, ReLU activation, BatchNorm and Dropout = 0.2 each.

**Architecture ResNet-50**: $\sim$ 23.5M params

Exactly the ResNet-50 architecture with addition of an input and an output layer.

*Input*: Batch (100 element) of 32x32x1 MNIST data (pad = 4).

*Input layer*: Covolutional layer with 64 channels, stride = 2, padding = 3, kernel size = 7.

*Output layer*: Linear layer with 10 output classes in order to perform classification, with bias.

*Optimizer*: SGD optimizer with learning rate of 1e-3 and momentum = 0.9.

**Hyperparameter search.** The relevant hyperparameters are $\beta$ and latent dimension $d$. $\mathcal{S}$-GVAE had better validation loss with $\beta = 1$ and $d = 10$, but $\beta = 0.1$ and $d = 6$ results in better disentanglement and reconstruction (Table 2). This is resonable since a high latent dimension reduces information loss but hurts generalization. High $\beta$ prioritizes $KL$ term over reconstruction.

Table 2: Validation loss on various combinations of $\beta$ and latent dimension after 50 epochs for a $\mathcal{S}$-GVAE and a $\mathcal{S}$-VAE (in parenthesis).

| $\beta$: | 0.01 | 0.1 | 1 | 150 | 500 |
|---|---|---|---|---|---|
| d=3 | 43.4k (37.8k) | 42.6k (38.3k) | 43.1k (37.4k) | 43.4k (38.4k) | 44.5k (41.7k) |
| d=6 | 30.3k (21.9k) | 30.9k (21.9k) | 30.0k (22.2k) | 32.9k (24.9k) | 38.0k (29.8k) |
| d=10 | 48.3k (13.4k) | **24.5k** (13.5k) | 24.7k (**13.2k**) | 28.5k (18.1k) | 35.5k (25.4k) |

We retrained top-performing models for 100 epochs, resulting in final validation losses of 30.3k and 13.4k respectively.

