# OpenReview forum: "Learning Rotation-Agnostic Representations via Group Equivariant VAEs"
_ICLR.cc/2023/TinyPapers — Submitted to Tiny Papers @ ICLR 2023_

### Official Review · Reviewer_DanT · 2023-03-20

**Confidence:** 2

**Summary Of Contributions:**

This paper aims to disentangle image representations from rotations through the use of a group-equivariant neural network (G-CNN). It shows that image embedding agnostic to rotations can be naturally obtained by implementing directly a SO(2) equivariant CNN as the encoder of a S-VAE.

**Rating:**

Clear, Correct, and Reproducible (CCR): a submission which meets the reviewing criteria

**Strengths And Weaknesses:**

Strengths
- The paper introduces its objective clearly.
- The methodology and experiment are described in detail.

Weaknesses
- There is no comparison between the proposed VAE with G-CNN encoder and VAE without G-CNN encoder.
- The paper doesn’t try other G-CNN architectures or other VAE architectures.

**Suggested Changes:**

- Adding more evaluation metrics aside from loss could better represent the results.
- If there are some relevant previous works, it would be better to include them in the experimental result to show how the proposed method fares against them.
- Adding more datasets to experiment on would also show how the proposed method generalizes across tasks.

---

### Official Review · Reviewer_AZv4 · 2023-03-22

**Confidence:** 3

**Summary Of Contributions:**

This paper proposes to add group equivariance to the encoder of a hyperspherical variational auto-encoder to disentangle group transformations from image content. Results on disentangling rotation from an augmented MNIST dataset with hyperparameter search show the potential of this method.

**Rating:**

Great Start (GS): a submission which meets some of the reviewing criteria but has room for improvement

**Strengths And Weaknesses:**

Strengths:
* Searching across multiple hyperparameter values strengthens the results, and provide evidence for the claims/analysis of the roles of these hyperparameters.
* Experimental details are provided in Appendix A, and it seems that code will be shared after de-anonymization.
* The paper is grammatically well-written overall.


Weaknesses:
* Discussion of relevant literature could be improved.
* The results are not presented very clearly, and need further analysis to be fully developed.

**Suggested Changes:**

The results are not very clear nor thoroughly analyzed. After explaining explicitly and/or labeling the top and bottom portions of Figure 1, what can we conclude from them? Why are the disentangled representations mostly properly oriented? Why are validation loss results presented in Table 1, when the hyperparameters were finally selected by performance in disentanglement and reconstruction? How are these latter qualities measured?

Some citations could be improved. What is the "β" of β-VAE? Davidson et al. (2018) should be cited immediately after naming S-VAEs. What is the comparison between your work and Vadgama et al. (2022)?

Sharvaree Vadgama, Jakub Mikolaj Tomczak, and Erik J. Bekkers. Kendall Shape-VAE: Learning Shapes in a Generative Framework. NeurIPS 2022 Workshop on Symmetry and Geometry in Neural Representations. 2022.

---

### Author Response · Authors · 2023-05-31
**Opt-in archival preference**

We confirm we would like to opt-in for archival on DBLP.

---

### Meta-Review · Area_Chair_7ruJ · 2023-04-07

**Recommendation:** Invite to archive
**Confidence:** 5

**Metareview:**

In this study, the authors aim to disentangle image content from its rotation, which is particularly desirable in various applications such as astronomical data and histopathological images. To achieve this goal, they propose the use of a Group Equivariant Convolutional Neural Network (G-CNN) encoder and demonstrate its effectiveness on the MNIST dataset.


**Summary:**

This paper aims to disentangle image representations from rotations through the use of a group-equivariant neural network (G-CNN).

**Reason For Not Giving A Higher Recommendation:**

This paper is  well-motivated with a clear and logical structure that makes it easy to follow and outlines the problem of rotational disentanglement . The authors provide a good introduction and  show the effectiveness of the proposed methodology in disentangling the content of images from their rotation on the MNIST dataset.

However, I concur with the reviewers that the current version of the paper requires significant changes to ensure it is Clear, Correct, and Reproducible (CCR). I believe the paper is close to achieving this and could benefit from the following modifications:

1. The presentation of the paper and specifically the result section needs improvement, and a more in-depth analysis is required to fully understand the implications of the findings. As it stands, the results are not sufficiently clear.

2. A comparison between the proposed VAE with the G-CNN encoder and a VAE without the G-CNN encoder is essential to establish the validity of the approach. Without this comparison, the paper's impact may be limited, as readers may not be convinced of the advantages of the proposed method. It is important to note that even if a VAE without a G-CNN encoder performs better or similarly, this comparison is crucial to provide a comprehensive evaluation of the proposed technique.


**Reason For Not Giving A Lower Recommendation:**

N/A

---

> ### Author Response · Authors · 2023-05-31
> **Thanks for the insightful review and constructive feedback**
>
> Dear Reviewers,
>
> Thank you for your insightful review and constructive feedback. We concur with your suggestions and have undertaken a revision of our manuscript to address the highlighted concerns.
>
> We've enriched the Experiments section by providing a more in-depth analysis of our findings. We've made concerted efforts to articulate our results more clearly, elucidating the implications of our research to give readers a fuller understanding.
>
> Furthermore, as per your suggestion, we performed a comparative analysis of the Variational Autoencoder (VAE) equipped with a Group Equivariant Convolutional Neural Network (G-CNN) encoder against a baseline VAE without a G-CNN encoder. These results have been included in the revised manuscript. This comparative evaluation has yielded informative insights, further validating the efficacy of our proposed method. We also added a deep dive on the classification of 6s and 9s digits, to better understand the implications of an architecture agnostic to rotations.
>
> We enriched the appendix with all the details needed to reproduce the experiments and added a link to the code used to perform the experiments (published as a notebook).
>
> We appreciate your thorough review and believe that these modifications significantly enhance our manuscript's clarity, correctness, and reproducibility.
>
> Sincerely,
>
> Paper223 authors

---

### Decision · Program_Chairs · 2023-04-09

Invite to archive